# Perspective of Internet Poker Players on Harm-Reduction Strategies: A Cross-Sectional Study

**DOI:** 10.3390/ijerph17239054

**Published:** 2020-12-04

**Authors:** Patrycja Michalska, Anne Chatton, Louise Penzenstadler, Paweł Izdebski, Emilien Jeannot, Olivier Simon, Magali Dufour, Lucien Rochat, Suzanne Lischer, Yasser Khazaal

**Affiliations:** 1Faculty of Psychology, Kazimierz Wielki University, 85-867 Bydgoszcz, Poland; Patmic33@wp.pl (P.M.); pawel@ukw.edu.pl (P.I.); Yasser.khazaal@chuv.ch (Y.K.); 2Department of Psychiatry, Geneva University Hospital, 1205 Geneva, Switzerland; anne.chatton@hcuge.ch (A.C.); louise.penzenstadler@hcuge.ch (L.P.); Lucien.rochat@unige.ch (L.R.); 3Institute of Global Health, Geneva University, 1211 Geneva, Switzerland; 4Addiction Medicine, Department of Psychiatry, Lausanne University Hospital, 1011 Lausanne, Switzerland; Olivier.simon@chuv.ch; 5Faculty of Biology and Medicine, Lausanne University, 1005 Lausanne, Switzerland; 6Department of Psychology, Université du Québec à Montréal, Montreal, CP 8888, Canada; dufour.magali@uqam.ca; 7Institute for Social Management, Social Policy and Prevention, Lucerne University of Applied Sciences and Arts, 6002 Lucerne, Switzerland; Suzanne.lischer@hslu.ch; 8Research Center, Montreal University Institute of Mental Health, Montreal, QC H3C 3P8, Canada

**Keywords:** online gambling, prevention strategies, responsible gambling, harm reduction, Internet gamblers, poker

## Abstract

*Background:* Internet gambling may increase rates of gambling harm. This current study aimed to assess Internet poker players’ views on various harm-reduction (HR) strategies. It also examined differences in these views according to the games played (poker only vs. poker plus other gambling activities), indebtedness, and problem gambling severity. *Methods:* Internet poker players (*n* = 311; 94.2% Male) recruited online between 2012 and 2014 were included in the analyses and completed a survey on indebtedness, problem gambling severity index, and ten statements regarding HR features. *Results:* Among the whole sample, the most frequently endorsed HR strategy was setting money limits, specialized online help, and peer support forums. People who play poker only (70%) are less prone to endorse the utility of information on excessive gambling and specialized healthcare centers. No differences were found between those people with debt versus those without regarding HR assessment. Participants with severe problem gambling were more skeptical about HR strategies based on information on specialized healthcare centers. *Conclusion:* Setting money limits, online help, and peer support forums are the most commonly endorsed strategies. Future research is needed to evaluate the effectiveness of online harm reduction strategies.

## 1. Introduction

### 1.1. Poker Characteristics

Poker is a competitive social gambling card game of skill and luck. The game includes a wide range of challenging strategic and interpersonal choices in a context of risk and uncertainty [1]. In the last few years, poker has turned into a multi-million-dollar industry [2].

Due to its unique game characteristics and the players‘ psychological and behavioral specificity, poker can be considered different from other gambling games [3]. Research with land-based gamblers has found that poker players and casino gamblers differ on measures of novelty seeking and gambling problems which were found to be higher and more frequent in the former than in the latter [4]. Furthermore, in a game of skill such as poker, players may overestimate their capacity to win (perceived skills) while relating their loss to bad luck [5]. Therefore, poker activity challenges existing theoretical concepts about problem gambling behaviors [3].

High rates of problem gambling and indebtedness were repeatedly reported among on-line gamblers [6]. However, using a simple dichotomy of online versus offline gamblers to examine the impact of Internet gambling seems inadequate according to some authors [7]. One may speculate that a broader taxonomy of online poker gamblers may exist ranging from those who play poker only to those who play poker and other games. With regard to problematic gambling, previous studies of poker players have identified the number of gambling activities as contributing to gambling problems [8,9]. 

### 1.2. Harm Reduction Strategies

Even if the group of online poker players is not homogeneous, it is crucial to develop customized harm reduction (HR) features that target this specific group of gamblers. HR-initiatives and programs are designed to reduce the magnitude of gambling-related harm by assisting gamblers with various strategies including helping gamblers to maintain their gambling expenditure within affordable limits [10]. There is currently a large variation in the type and extent of responsible gambling and HR features for Internet gambling [11,12]. Effective interventions are crucial to assist gamblers at various levels of risk. They can allow them to acquire and apply the requisite skills and knowledge in order to maintain their gambling within affordable limits [13].

Since the main goal of a HR program is to prevent gambling-related harms, programs should inter alia provide information that consumers could use to make decisions [14]. Thus, gamblers should receive: (a) information about the dangers of excessive gambling and how to avoid them; (b) resources for help-seeking; (c) information about how games really work, and; (d) education regarding common misconceptions that encourage false beliefs about the probabilities of winning [15]. Nevertheless, some mixed results showed that information [16] and awareness campaigns hardly modify irrational beliefs or gambling behavior. 

One of the most widely used public health strategies for reducing harm from gambling is prevention messages. Messaging seeks to elicit direct changes in behaviors/beliefs and may inform the public of associated risks or programs. With recent technological changes in gambling products, it is hypothesized that customized messages may be more effective than a one-size-fits-all messaging program. Tailoring the messages to specific player groups could increase their acceptability (i.e., skilled-game gamblers prefer more direct communication) [13]. Internet gambling allows delivering in-game tailored warning signs for prolonged play or high expenditure, including possible player behavioral tracking and personalized feedback as well as alerts (warning messages) during the game sessions [17,18]. In spite of possible negative perception by some gamblers (disruption of game enjoyment) [17], such messages are expected to be potentially helpful. 

One of the more widespread types of social responsibility tools concerns limits setting [19]. The goal of this tool is to enable gamblers to pre-set monetary and time limits in a non-emotional state to assist them in spending only as much as they can afford to lose. Depending on the gaming venue or website, spending limits can include deposit, play, loss, win and bet limits. Time limits can be provided for a game session within daily, weekly, and monthly time frames [20]. However, there is mixed evidence on the impact of deposit limits for online gambling; some studies indicate reduced spending as a result, others show small or no effects [21]. Moreover, not all providers offer this option [1]. Overall, a distinction has to be made between voluntary and mandatory game limits. Voluntary limit setting tools allow players to pre-set the amount of time and/or money they wish to spend on gambling in a specified time period (typically per day and/or per calendar month) [22]. On the other hand, mandatory (i.e., operator-imposed) limits can include time and/or money spending thresholds (gambling opportunities are automatically blocked upon the reach of such thresholds) [23]. However, according to one study, operator-imposed limits are not always appropriate for at least some of the players [24]. 

### 1.3. Gamblers’ Attitudes towards Harm Reduction Strategies

Several studies have examined players’ attitudes towards limit-setting tools [13,22,25]. An evaluation of 50 online gambling sites revealed that monetary limit-setting tools are generally positively viewed because they encourage gamblers to reflect on the amount of time they spend gambling [26]. A recent study stressed that players were generally aware of the consumer protection tool and had accessed activity statements; even though few had used deposit limits or time-outs, the use of these restrictive tools was higher among those at-risk of gambling problems [21]. 

It seems that gamblers generally support the availability of HR tools, particularly those that assist customers to play within their means, including player feedback and regular financial statements. Customer engagement with HR tools, however, appears to be relatively low [13]. 

The Internet is not only changing gambling, but it also provides new opportunities for more interactive and in-game HR tools. It is therefore of particular interest to understand how Internet gamblers, and more specifically the emerging subgroup of Internet poker gamblers, consider the potential efficacy of such tools. 

### 1.4. Aims

This research aims thus to preliminarily explore online poker gamblers’ attitudes concerning HR measures. Besides, the study focuses on the question of whether gamblers’ attitudes differ across groups with respect to the gamblers’ type of game (poker only versus poker plus other gambling activities), indebtedness, and the severity of problem gambling, with the hypothesis that such characteristics influence poker gamblers’ views on HR tools.

## 2. Materials and Methods

### 2.1. Participants and Procedure

The current study was a secondary analysis of a previously published study which aimed to identify latent groups of online gamblers from a number of social-related variables [27]. Adult Internet gamblers were recruited through advertisements posted in specialized poker dedicated forums and websites between January 2012 and March 2014. The questionnaires were completed anonymously online after obtaining informed consent online, without any compensation. The study was conducted in accordance with the Declaration of Helsinki, and the protocol was approved by the Ethical Committee of the Geneva University Hospitals.

Of the 584 participants who took part in the survey, 311 online poker gamblers were retrieved for the purposes of this study and analyzed. The others were excluded because either they were off-line gamblers or not poker gamblers.

### 2.2. Measures

Participants were asked to give information on their demographics (age, gender) and gambling behaviors including the game type which allowed a comparison between pure poker players and mixed gamblers. It was, for instance, reported that gamblers who play skills and luck games had higher gambling severity scores and higher coping motivation for gambling than skill gamblers [28]. They also completed assessments of possible gambling-related debts in the past 12 months (adapted from the South Oaks Gambling Screen [29]) as well as the following measures:

#### 2.2.1. The Problem Gambling Severity Index

Problem Gambling Severity Index (PGSI) was a nine-item measure of the severity of gambling problems. The scoring allowed a classification of individuals according to gambling risk: (0) no risk; (1–2) low risk; (3–7) moderate risk; and (8 through higher) problem gamblers [30]. 

#### 2.2.2. Harm Reduction Items

The participants were asked about their attitude towards various HR measures. They were specifically asked about their perception of whether the measures could be helpful in reducing gambling-related harm. For this purpose, a set of various preventive measures offered by providers were put together. The participants responded to ten statements regarding gambling prevention strategies on a 5-points Likert scale: (1) this would not help at all, (2) this would be slightly helpful, (3) this would be moderately helpful, (4) this would be very helpful, (5) this would extremely helpful.

## 3. Data Analysis

In this study, we used an SPSS 25.0 (IBM, Chicago, IL, USA) software program. Descriptive statistics such as mean (SD), median (min, max), or percentages were reported for demographics and gambling characteristics in Table 1. We analyzed harm reduction strategies according to game type (Table 2), indebtedness (Table 3), and the problem gambling severity index (Table 4). Due to the small sample size, the five original categories of the HR items were reduced to three (items 2 and 3 respectively, 4 and 5 were collapsed) in order to increase statistical power and further validate statistical inferences. *T*-tests (or the median test when required) and Chi-square tests of homogeneity were performed to test for group differences. Significance was set at *p* = 0.05.

Because multiple statistical tests were performed within the same analysis, we adjusted the raw *p*-value by the number of tests to be conducted in order to maintain the overall Type I error rate. This was done using the Bonferroni correction method where the rejection criterion value of 0.05 was divided by the respective number of tests performed in each table. The new critical *p*-values were, therefore, rounded to the nearest two significant digits, 0.0038, 0.0045, and 0.0050 in Table 2, Table 3 and Table 4 respectively. The *p*-value in bold indicates that the null hypothesis had been rejected after the multiple-test correction was made.

Finally, when the omnibus Chi-square test was significant, we used adjusted standardized residuals, following a Bonferroni adjustment, and the ones exceeding 1.96 in absolute value helped identify those specific cells making the greatest contribution to the omnibus Chi-square result. If the criteria of +/− 1.96 were met, this would indicate that there were more/fewer participants in a condition than would be expected by chance [31,32].

### Handling of Missing Data

As PGSI scores were missing in almost 18% of the cases, participants with missing observations on this variable were compared with those with complete data against selected gambling characteristics such as the time since gambling began, the number of hours spent gambling, indebtedness due to gambling, and source of income to make sure that no systematic differences existed between the two groups. 

These missing observations explained why the totals did not match in the different tables.

## 4. Results

The sample included mostly men (94.2%) and the average age was 30.6 years (SD = 8.1). About 70% of the participants exclusively played poker online whereas the others were also involved in different kinds (one or more) of Internet gambling activities (e.g., 26.6% in sports betting, 6.2% in slot machines, 5.1% in lottery games, 9.7% in casino games, 7% other games) besides poker (data not shown). Other demographics and gambling characteristics of the sample are reported in Table 1.

When considering the whole sample in Table 2, only two HR measures were endorsed (the last category “would be very or extremely helpful” is considered as an endorsement) by at least 50% of the study participants: setting money limits (53.4%) and specialized online help (50.0%). Another kind of online help, peer support forums with other gamblers, was endorsed by 44% of the sample. Two of the items were endorsed by less than 30% of the participants: information on excessive gambling and its consequences (25.2%) and warnings based on playing time (28.3%).

Finally, the analyses of participants with missing data on the PGSI compared with those with complete observations showed no systematic difference between the two groups (output not shown).

### 4.1. Harm Reduction Items by Game Type

Table 2 shows the distribution of the gambling severity score, indebtedness status, and participants’ attitudes towards prevention measures by game type. The median test showed no statistically significant between-group difference regarding their gambling severity (Med = 2 and *p* = 0.5). However, a statistical between-group difference was observed for indebtedness: those playing online poker plus other games compared to online poker only were significantly more indebted (15.7% vs. 5.0%, χ^2^
_(1)_ = 9.8 and *p* = 0.002). As far as the HR measures were concerned, there were significant statistical differences between online poker only and online poker plus other games in their answers to the first two preventive measures (χ^2^
_(2)_ = 14, *p* = 0.001 and χ^2^
_(2)_ = 10.4, *p*= 0.006 respectively). Adjusted standardized residuals showed that responses 1 and 2 were the ones that significantly contributed to the omnibus Chi-square result. Indeed, there were more participants in the poker only condition which rated that HR1 “information on excessive gambling and its consequences” would be moderately or slightly helpful compared to the other conditions (68.4% vs. 43.3%), whereas people playing online poker plus other games were more likely than the other group to endorse this proposition (40.3% vs. 19.3). The same distribution was observed regarding the HR2 measure on specialized healthcare centers: 60.0% vs. 37.3% were skeptical about this proposal while 50.7% vs. 34.1% endorsed it. Group membership did not differentiate the other prevention items.

### 4.2. Harm Reduction Items by Indebtedness Status

In Table 3, women represented less than 8% of indebted players. Regarding the PGSI score, the median test was statistically significant between the indebted and the non-indebted players (*p* < 0.001). It showed a wider range in the indebted group compared to the non-indebted one [(0–27) vs. (0–15)]. Group membership differentiated none of the preventive measures.

### 4.3. Harm Reduction Items by Problem Gambling Severity Index (PGSI)

Table 4 shows how HR measures were distributed according to the participants’ PGSI severity index. There was a significant between-group difference regarding indebtedness (χ^2^
_(2)_ = 73.4, *p* < 0.001). Adjusted standardized residuals showed that people with moderate problem gambling were more likely to be indebted than those with low or without problem gambling (8.8% vs. 0.7%). Similarly, people with severe problem gambling tended to be overrepresented in the indebted group when compared to those with moderate or no problem gambling (60% vs. 8.8% respectively 60% vs. 0.7%). As far as the HR measures were concerned, there was a significant group difference (χ^2^
_(4)_ = 9.8, *p* = 0.04) for the second one only (information on specialized healthcare centers). Adjusted standardized residuals showed that people with severe problem gambling compared to those without or low levels of problem gambling (26.7% vs. 5.3) were more skeptical (*would not help at all*) about this proposal. There were no other significant differences.

## 5. Discussion

The aim of this study was to examine online poker gamblers’ attitudes concerning HR measures. Besides, the study examined whether gamblers’ attitudes differ across groups related to the type of game (poker only versus poker plus other gambling activities), and to what extent the attitude is influenced by the degree of indebtedness and gambling-related problems.

First, when considering all participants, the proposed strategies were endorsed to some extent varying from 25% to 53% of the participants. None of the HR measures were completely rejected by the participants. Three of them received around 50% endorsement rates, namely, setting money limits, automatic block after losing or wagering a certain amount of money (mandatory limit), and specialized online help. 

Given that playing poker is a more time-consuming game than almost all other forms of gambling, it could be argued that voluntary limit setting impacting on the duration of play would be a desirable outcome of limit-setting for this particular type of player [33]. Nonetheless, setting time limits, as shown in other studies [34], was less endorsed than monetary limits [34]. An automatic block when mandatory money limits were wagered and/or lost was considered a helpful strategy by almost 50% of the sample. Messages and warnings related to money limits were also endorsed by about 45% of the sample indicating that Internet poker gamblers consider such interactive in-game tracking and interventions [35] as possibly helpful. Endorsement of online help has to be considered with specific attention as it requires the development of Internet interventions dedicated to gambling problems [36,37,38,39,40]. Internet-delivered treatments for gambling problems could be particularly helpful to fill the treatment gap faced by people with gambling disorders [41]. Internet-based interventions could have very different forms such as web format [38,42], smartphone apps [40,43], or specialized email or video call support. Future studies may further assess different kinds of assistance, their acceptability, usability, and effectiveness. The results at hand, however, at least argue for the indication of some online HR support on the Internet by gamblingoperators. Finally, in consideration of the important comorbidity between gambling disorders and other psychiatric disorders [44,45], possible transdiagnostic interventions (i.e., focusing on emotion regulation) [46] might also be helpful. 

A substantial proportion (50%) of the participants also endorsed online peer support as a helpful tool. Online gambling communities are widely used by gamblers and specifically by people with pathological gambling disorders [47]. However, the gambling communities seem rather orientated towards sharing gambling tips than on HR or recovery. Therefore, it was supposed to increase the risk of gambling problems [47] by maintaining gambling fallacy rather than increasing control over gambling [48]. Nevertheless, it seems, according to the present results, that peer support is also viewed by poker players as a helpful strategy as reported for other addictive disorders [37].

Second, people who play poker plus other games, in comparison to those playing poker only, more frequently endorse items related to psychoeducation (information on gambling-related risks and on specialized healthcare centers). They are also more frequently indebted and have a higher score on the PGSI. It seems that those participants are at higher risk for such problems and value more such kind of help. One may hypothesize that information on specialized healthcare centers would include not only local addresses but possibly also information on how such centers could help reduce stigma and shame associated with gambling treatment venues. In addition, indebted participants more frequently endorsed items related to specialized help (clinical centers and online help). Similarly, the group with the greatest problem gambling level endorsed more frequently than the other poker players the item related to information on specialized healthcare. It seems that gamblers facing problems consider the need for specialized interventions. However, looking at the whole sample, a large part of the participants were skeptical regarding the efficacy of the proposed interventions. Similar findings were reported in another study showing that less than half of the sample considered that monetary-loss limits could be helpful [22]. The finding is possibly influenced by public stigma related to gambling disorders [49], lack of knowledge about possible HR strategies, lack of metacognitive capacity to question possible biases or irrational beliefs or previous unsuccessful experiences. Acceptability and effectiveness, as well as the availability of the different tools, should be assessed and repetitively improved taking into account users’ views and needs. 

In spite of concerns related to the specific needs of female gamblers [50,51], it is difficult to draw any conclusion on gender-specific needs from the data at hand due to the small number of women involved in the study. In comparison to men, women are possibly more prone to consider online gambling as safer, less intimidating, anonymous, and more acceptable than places like a physical casino, bar, or race track [52]. Previously, problem gambling in women was associated with a feeling of loneliness [27]. In consideration of such findings, specific views on the needs of female Internet gamblers should be further assessed.

## 6. Limits of the Study

The results of the study should be read with several limitations in mind, particularly its cross-sectional design. Participants were recruited from the Internet and may gamble on poker from different websites and different countries with various policies and harm reduction tools. This study did not include all possible strategies. For instance, in consideration of the high number of available websites for gambling, a self-exclusion strategy was not included [53]. Furthermore, the proposed strategies are mostly related to what players can do themselves to prevent gambling-related risks. Such HR strategies are in coherence with the so-called “responsible gambling perspective” [54,55]. This perspective is, however, criticized because it focuses on the gambler’s responsibility only, without taking into account structural aspects of the games as well as gambling products and commercial practices [56,57]. 

In addition, when comparing the participants according to the three characteristics (poker only versus poker plus other games, indebtedness, gambling severity), only a few significant differences emerged across the groups, which might be due to the small sample size in some groups and the related lack of power. The study is based on self-assessment methods and lacks details about gambling frequency, however, weekly hours spent on Internet gambling as well as PGSI are assessed. Furthermore, due to the online recruitment method, the study was subjected to self-selection biases [58]. Finally, further studies may assess Internet gamblers’ use of the different harm-reduction strategies in real-time using methods like behavioral tracking data analyses.

## 7. Conclusions

Information-related strategies were rarely endorsed by poker gamblers in comparison to more structural ones. The possible efficacy of some of the tools could be also influenced by other factors such as impulsivity [59]. Future research is needed to evaluate the effectiveness of HR strategies, including limit-settings, with the aim of increasing the effectiveness and acceptability of such interventions in representative samples of gamblers [20,60]. HR strategies may also benefit from perspectives acknowledging the socio-cultural influences on Internet-gambling related harm [61]. Involving users in the development and assessment of the results highlights the importance of involving users in the assessment of HR strategies. This finding is in accordance with calls and studies in the psychiatric and addictive disorders fields [62,63] showing the importance of user-involvement approaches. Involving Internet gamblers in HR conception and assessment could be by itself a dynamic and continuous structural HR strategy. 

Most of the HR strategies reported in the present study were endorsed by the participants across different levels of gambling severity suggesting possible usefulness also among the lower risk group of gamblers. The effectiveness of such interventions is however limited by the extent to which gamblers are prone to use the systems when they need it. 

HR strategies could be further improved by better awareness of emotional reactivity during game sessions and their possible impact on players’ loss of control [64]. In addition to effective HR tools, other preventative population approaches, such as supply reduction or less addictive gambling design, are needed to reduce the global burden of problem Internet gambling [12].

## Figures and Tables

**Table 1 ijerph-17-09054-t001:** Socio-demographics and gambling characteristics of the participants. Data are expressed as mean (±SD), median (range), or percentage.

Sample main characteristics ccc	*n* = 311
Age	30.6 (8.1)
Sex	
- male	94.2
Living status: I live	
- alone	27.1
- in a relationship	31.3
- with my family	32.3
- with friends	2.9
- with roommates	6.5
Annual income (in euros)	
- less than 30,000	53.4
- between 30,000 and 50,000	22.0
- more than 50,000	24.6
I have been gambling on the Internet since:	
- less than 2 years	16.1
- between 2 and 5 years	47.6
- more than 5 years	36.3
Number of hours gambling in a week on the Internet	15.0 (1–50)
Game types *	
- poker	99.7
- sports betting	19.0
- casino	7.1
- lottery	3.9
- slot machines	3.5
- scratching games	3.2
- other	1.9
Number of gaming sites visited in the last 3 months	
- only one	42.2
- more than one	57.8
In the last 12 months gambling has been:	
- my main source of income	14.5
- an ancillary source of income	30.2
- not a significant source of income	55.3

* Total exceeds 100% due to multiple responses.

**Table 2 ijerph-17-09054-t002:** Harm reduction items by game type. Data are expressed as mean (±SD) or percentage.

	All (*n* = 311)	Online Poker Only (*n* = 222)	Online Poker Plus Other Games (*n* = 89)	*p*-Value ^a^
Age	30.6 (8.1)	30.0 (7.8)	32.2 (8.7)	0.3
Sex -male	94.2	94.5	93.2	0.6
Did you borrow money in the past twelve months to gamble or to pay for gambling debts?				**0.002**
- yes	8.0	5.0	15.0	
1. Information on excessive gambling and its consequences:				**0.001**
- this would not help at all	13.4	12.3	16.4	
- this would be slightly or moderately helpful	61.3	68.4	43.3	
- this would be very or extremely helpful	25.2	19.3	40.3	
2. Information on specialized healthcare centers:				0.006
- this would not help at all	7.6	5.9	11.9	
- this would be slightly or moderately helpful	53.6	60.0	37.3	
- this would be very or extremely helpful	38.8	34.1	50.7	
3.Setting individual time limits on Internet gambling sites:				0.3
- this would not help at all	14.7	16.4	10.4	
- this would be slightly or moderately helpful	45.4	46.2	43.3	
- this would be very or extremely helpful	39.9	37.4	46.3	
4. Messages and warnings based on playing time limits:				1.0
- this would not help at all	15.6	15.2	16.7	
- this would be slightly or moderately helpful	56.1	56.1	56.1	
- this would be very or extremely helpful	28.3	28.7	27.3	
5. Gambling opportunities are automatically blocked after a set time (mandatory limit):				0.7
- this would not help at all	19.7	21.1	16.4	
- this would be slightly or moderately helpful	39.1	39.2	38.8	
- this would be very or extremely helpful	41.2	39.8	44.8	
6. Setting individual money limits, for instance, players can set their own money limit to gamble and to lose on their profile:				0.4
- this would not help at all	5.5	4.7	7.6	
- this would be slightly or moderately helpful	41.1	43.5	27.1	
- this would be very or extremely helpful	53.4	51.8	57.6	
7.Messages and warnings concerning the total amount wagered and lost:				0.5
- this would not help at all	9.7	8.2	13.4	
- this would be slightly or moderately helpful	45.4	45.6	44.8	
- this would be very or extremely helpful	45.0	46.2	41.8	
8.Gambling options are automatically blocked after a certain amount wagered and lost (mandatory limit):				0.6
- this would not help at all	11.1	11.2	10.6	
- this would be slightly or moderately helpful	39.6	41.1	34.8	
- this would be very or extremely helpful	49.4	47.3	54.5	
9.Offering online forums for discussion with other gamblers:				0.07
- this would not help at all	8.0	5.8	13.4	
- this would be slightly or moderately helpful	47.9	51.5	38.8	
- this would be very or extremely helpful	44.1	42.7	47.8	
10.Online support with a specialist on a website dedicated to gambling problems				0.3
- this would not help at all	7.1	5.8	10.4	
- this would be slightly or moderately helpful	42.9	45.6	35.8	
- this would be very or extremely helpful	50.0	48.5	53.7	

^a^*p*-values result from Chi-square tests of homogeneity for categorical variables and T-tests for continuous variables respectively.

**Table 3 ijerph-17-09054-t003:** Harm reduction items by indebtedness status. Data are expressed as a percentage.

	Indebted Players (*n* = 25)	Non Indebted Players (*n* = 284)	*p*-Value ^a^
1.Information on excessive gambling and its consequences:			0.8
- this would not help at all	11.1	13.8	
- this would be slightly or moderately helpful	55.6	61.5	
- this would be very or extremely helpful	33.3	24.8	
2. Information on specialized healthcare centres:			0.1
- this would not help at all	16.7	6.9	
-this would be slightly or moderately helpful	33.3	54.8	
- this would be very or extremely helpful	50.0	38.2	
3. Setting individual time limits on Internet gambling sites:			1.0
- this would not help at all	16.7	14.7	
- this would be slightly or moderately helpful	44.4	45.4	
- this would be very or extremely helpful	38.9	39.9	
4. Messages and warnings based on playing time limits:			0.7
- this would not help at all	22.2	15.2	
- this would be slightly or moderately helpful	50.0	56.2	
- this would be very or extremely helpful	27.8	28.6	
5. Gambling opportunities are automatically blocked after a set time:			0.9
- this would not help at all	22.2	19.7	
- this would be slightly or moderately helpful	33.3	39.4	
- this would be very or extremely helpful	44.4	40.8	
6. Setting individual money limits, for instance, players can set their own money limit to gamble and to lose on their profile:			0.7
- this would not help at all	5.6	5.6	
- this would be slightly or moderately helpful	50.0	40.3	
- this would be very or extremely helpful	44.4	54.2	
7. Messages and warnings concerning the total amount wagered and lost:			1.0
- this would not help at all	11.1	9.6	
- this would be slightly or moderately helpful	44.4	45.0	
- this would be very or extremely helpful	44.4	45.4	
8. Gambling options are automatically blocked after a certain amount wagered and lost:			0.3
- this would not help at all	0	12.1	
- this would be slightly or moderately helpful	50.0	38.6	
- this would be very or extremely helpful	50.0	49.3	
9. Offering online forums for discussion with other distressed gamblers:			0.9
- this would not help at all	5.6	8.3	
- this would be slightly or moderately helpful	44.4	47.7	
- this would be very or extremely helpful	50.0	44.0	
10.Online support, with a specialist on a website dedicated to gambling problems			0.4
- this would not help at all	5.6	7.3	
- this would be slightly or moderately helpful	27.8	43.6	
- this would be very or extremely helpful	66.7	49.1	

^a^*p*-values result from Chi-square tests of homogeneity for categorical variables.

**Table 4 ijerph-17-09054-t004:** Harm reduction items according to the Problem Gambling Severity Index (PGSI). Data are expressed as a percentage.

	Non-Problem to Low Level of Problem Gambling (*n* = 148)	Moderate Level of Problem Gambling (*n* = 93)	High Level of Problem Gambling (*n* = 15)	*p*-Value ^a^
1. Information on excessive gambling and its consequences:				0.2
- this would not help at all	12.8	12.2	26.7	
- this would be slightly or moderately helpful	57.1	68.9	53.3	
- this would be very or extremely helpful	30.1	18.9	20.0	
2.Information on specialized healthcare centers:				0.04
- this would not help at all	5.3	7.8	26.7	
- this would be slightly or moderately helpful	53.8	56.7	33.3	
- this would be very or extremely helpful	40.9	35.6	40.0	
3. Setting individual time limits on Internet gambling sites:				0.6
- this would not help at all	17.3	12.2	6.7	
- this would be slightly or moderately helpful	46.5	43.3	46.7	
- this would be very or extremely helpful	36.1	44.4	46.7	
4. Messages and warnings based on playing time limits:				0.9
- this would not help at all	15.9	14.4	20.0	
- this would be slightly or moderately helpful	57.6	54.4	53.3	
- this would be very or extremely helpful	26.5	31.1	26.7	
5. Gambling opportunities are automatically blocked after a time set:				0.5
- this would not help at all	21.8	16.7	20.0	
- this be slightly or moderately helpful	41.4	37.8	26.7	
- this would be very or extremely helpful	36.8	45.6	53.3	
6. Setting individual money limits, for instance, players can set their own money limit to gamble and to lose on their profile:				0.5
- this would not help at all	6.1	3.3	13.3	
- this would be slightly or moderately helpful	40.5	42.2	40.0	
- this would be very or extremely helpful	53.4	54.4	46.7	
7. Messages and warnings concerning the total amount wagered and lost:				0.8
- this would not help at all	9.0	10.0	13.3	
- this would be slightly or moderately helpful	48.9	41.1	40.0	
- this would be very or extremely helpful	42.1	48.9	46.7	
8. Gambling options are automatically blocked after a certain amount wagered and lost:				0.9
- this would not help at all	11.4	11.2	7.1	
- this would be slightly or moderately helpful	41.7	37.1	35.7	
- this would be very or extremely helpful	47.0	51.7	57.1	
9. Offering online forums for discussion with other gamblers:				0.9
- this would not help at all				
- this would be slightly or moderately helpful	8.345.9	6.751.1	13.346.7	
- this would be very or extremely helpful	45.9	42.2	40.0	
10.Online support, with a specialist on a website dedicated to gambling problems				0.6
- this would not help at all	9.0	4.4	6.7	
- this would be slightly or moderately helpful	44.4	42.2	33.3	
- this would be very or extremely helpful	46.6	53.3	60.0	

^a^*p*-values result from Chi-square tests of homogeneity for categorical variables.

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
