# Peer review of "Perspective of Internet Poker Players on Harm-Reduction Strategies: A Cross-Sectional Study"

_ijerph, 2020, doi:10.3390/ijerph17239054_

Round 1
Reviewer 1 Report
The authors have addressed most of my comments except the first one:
I have doubt on the significance and conceptual implications of comparing poker only and poker plus other games players. The authors need to provide a convincing reason.
Author Response
Reviewer 1
Comments and Suggestions for Authors
Comment: The authors have addressed most of my comments except the first one:
I have doubt on the significance and conceptual implications of comparing poker only and poker plus other games players. The authors need to provide a convincing reason.
Answer: Thank you for your comment
The following modification (bold character) was added to the methods section:
Participants were asked to give information on their demographics (age, gender) and gambling behaviours including games type which allow comparison between pure poker players and mixed gamblers. It was for instance previously reported that gamblers who play skills and luck games had higher gambling severity scores and higher coping motivation for gambling than skill gamblers.
The following reference was added: Mathieu S, Barrault S, Brunault P, Varescon I. The role of gambling type on gambling motives, cognitive distortions, and gambling severity in gamblers recruited online. PloS one 2020;15:e0238978
In the submitted study we also found that:
“People who play poker plus other games, in comparison to those playing poker only, more frequently endorse items related to psychoeducation (information on gambling-related risks and on specialized healthcare centres). They are also more frequently indebted and have a higher score on the PGSI. “ Details of the distribution of the sample on the different games were added (table 1).
Reviewer 2 Report
Dear colleagues,
Thank you for giving me the opportunity of reading your work, it has been a very big pleasure to review it. The topic of this paper is one well-researched, but the quality of your work justifies the publication. In general, the paper is well-written. Despite this fact, I would suggest little changes to improve it:
- Regarding to the theoretical background, there are a lot of information. I think that it is possible to reduce a little bit the extension. Moreover, it will be useful to create subsections.
- There are some questions in the text that I don’t understand: for example, line 78 and 318.
- The hypothesis should appear at the end of introduction
- Your conclusions are very insightful, especially when you talk about limitations. However, I consider important to expand the limitations related to assessment tools to give an extra information to other researchers.
Author Response
Reviewer 2
Comments and Suggestions for Authors
Dear colleagues,
Comment: Thank you for giving me the opportunity of reading your work, it has been a very big pleasure to review it. The topic of this paper is one well-researched, but the quality of your work justifies the publication. In general, the paper is well-written.
Answer: Thank you for your comment
Despite this fact, I would suggest little changes to improve it:
- Comment: Regarding to the theoretical background, there are a lot of information. I think that it is possible to reduce a little bit the extension. Moreover, it will be useful to create subsections. Answer: Accordingly subsections were created
- Comment: There are some questions in the text that I don’t understand: for example, line 78 and 318. Answer: An appendix including the questions specifically made for the study was added
- Comment: The hypothesis should appear at the end of introduction. Answer: Accordingly, the aim of the study was modified as follows (bold characters): “This research aims thus to preliminary explore online poker gamblers’ attitudes concerning HR measures. Besides, the study focuses on the question whether gamblers’ attitudes differ across groups with respect to the gamblers’ type of game (poker only versus poker plus other gambling activities), indebtedness and the severity of problem gambling, with the hypothesis that such characteristics influence poker gamblers’ views on HR tools.
- Comment: Your conclusions are very insightful, especially when you talk about limitations. However, I consider important to expand the limitations related to assessment tools to give an extra information to other researchers. Answer: Accordingly, the following limitation was added to the discussion section “Finally, further studies may assess Inernet gamblers’ use of the different harm-reduction strategies in real-time using methods like behavioral tracking data analyses. »

Reviewer 3 Report
This paper addresses online poker players’ views on variety of measures which could manage them to gamble more responsibly such as setting time and money limits. I see a potential here, although there are a number of critical points and issues to consider before this paper is scientifically solid.
Abstract
-Please add more information into the Abstract for the readers.
For example, you could mention where and when (year/time frame) this study was conducted/ when the data were collected (began – closed). You could also mention the problem gambling measure used in this study. Please state also already in the abstract that this is a descriptive study of practices and/or the study was analyzed with the descriptive measures.
-The authors state that study “aims to assess Internet poker players’ views on various prevention strategies”. What do you mean by prevention strategies? This is a problematic term. Specify, give examples or change terminology in the abstract. I believe the authors assess views on various measures to cut potentially high-risk (excessive) gambling and/or assess practices to gamble more responsibly rather than prevent (all) gambling.
- I am also a bit uncomfortable with the first sentence in the abstract: “The majority of online poker players suffer from problem gambling”. I have a public health background and I am aware of a strong stigmatisation associated with use of this sentence. The authors may want to re-consider the sentence. You could say it in a more neutral way here such as that online gambling does pose an increased risk for problem gambling /high-risk playing or may increase rates of gambling harm etc. I believe that in general, online poker players are very heterogenous group and the sentence in its present format is somewhat questionable.
-Perhaps worth of mentioning in the abstract also is that majority of the study participants were men.
Methods:
Data procedure needs more details. I suggest that you inform readers more clearly about the process how the data were collected. For example, give some examples about the forums and websites from where the participants were recruited or were these websites gambling platforms or what? When the study was conducted? (year, time frame, began - closed). Please specify also questions used in this study. Did you asked about gambling frequency? If no, this is a severe mistake which needs discussion in the study limitation section. What about the game types asked? Is it possible to include the questionnaire/questions as an Appendix here?
Discussion: Study limitations needs more detail; the authors should consider at least limitations of the cross-sectional nature of the study and limitations related to the problem gambling measure used.
Terminology issues: Overall, I think that the terminology used in this paper is not uniform and is confusing. Sometimes the authors use the term “Internet poker” and sometimes (see e.g. Table 2) “online poker”. Sometimes the authors use the term RG measures and sometimes (gambling) prevention measures. There are several terminology-related issues to be re-considered. Also in the abstract, the authors use the term “gambling prevention strategies”. However, I think that most of the offered strategies were planned to reduce (or manage excessive) gambling, not prevent all gambling. Generally speaking, gambling prevention strategies include several types of strategies from societal level interventions to tools offered to individual gamblers etc..
Discussion: What about those people at risk of gambling problems? From the public health perspective, strategies for managing gambling and to prevent harm among this lower risk group of gamblers are also very important.
Author Response
Reviewer 3
Comments and Suggestions for Authors
This paper addresses online poker players’ views on variety of measures which could manage them to gamble more responsibly such as setting time and money limits. I see a potential here, although there are a number of critical points and issues to consider before this paper is scientifically solid.
Abstract
-Comment: Please add more information into the Abstract for the readers.
For example, you could mention where and when (year/time frame) this study was conducted/ when the data were collected (began – closed).
Starting collection date: January 2012
End date: March 2014
Answer: Accordingly added
Comment: You could also mention the problem gambling measure used in this study. Please state also already in the abstract that this is a descriptive study of practices and/or the study was analyzed with the descriptive measures. Answer: accordingly added
Comment: The authors state that study “aims to assess Internet poker players’ views on various prevention strategies”. What do you mean by prevention strategies? This is a problematic term. Specify, give examples or change terminology in the abstract. I believe the authors assess views on various measures to cut potentially high-risk (excessive) gambling and/or assess practices to gamble more responsibly rather than prevent (all) gambling Answer: Thank you for your comment. According to the introduction on Harm reduction and following your comments, we choose to replace the “prevention strategies” term by Harm-reduction, in the title , text and abstract.
Comment: I am also a bit uncomfortable with the first sentence in the abstract: “The majority of online poker players suffer from problem gambling”. I have a public health background and I am aware of a strong stigmatisation associated with use of this sentence. The authors may want to re-consider the sentence. You could say it in a more neutral way here such as that online gambling does pose an increased risk for problem gambling /high-risk playing or may increase rates of gambling harm etc. I believe that in general, online poker players are very heterogenous group and the sentence in its present format is somewhat questionable.Answer: Accordingly, this sentence was deleted and replaced by the following one: “Internet gambling may increase rates of gambling harm”
-Comment: Perhaps worth of mentioning in the abstract also is that majority of the study participants were men. Accordingly added in the abstract
Methods:
Comment: Data procedure needs more details. I suggest that you inform readers more clearly about the process how the data were collected. For example, give some examples about the forums and websites from where the participants were recruited or were these websites gambling platforms or what? Answer: “Adult Internet gamblers were recruited through advertisements posted in specialized Poker dedicated forums and websites”
Comment: When the study was conducted? (year, time frame, began - closed).
Answer: The time frame was accordingly added
Comment: Please specify also questions used in this study. Did you asked about gambling frequency? If no, this is a severe mistake which needs discussion in the study limitation section.
Answer: The survey asked about time since gambling, number of weekly hours spent on Internet gambling and assessed the gambling problem severity (with PGSI). Responses already provided in Table 1. Gambling frequency was however not assessed. A comment was accordingly added in the limitation section.
Comment: What about the game types asked?
Answer: Game types and number of gaming sites visited now assessed in Table 1 as follows:
Table 1. Socio-demographics and gambling characteristics of the participants. Data are expressed as mean (±SD), median (range) or percentage.
|
|
N=311 |
|
Age |
30.6 (8.1) |
|
Sex - male |
94.2 |
|
Living status: I live - alone - in a relationship - with my family - with friends - with roommates |
27.1 31.3 32.3 2.9 6.5 |
|
Annual income (in euros) - less than 30,000 - between 30,000 and 50,000 - more than 50,000 |
53.4 22.0 24.6 |
|
I've been gambling on the Internet since: - less than 2 years - between 2 and 5 years - more than 5 years |
16.1 47.6 36.3 |
|
Number of hours gambling in a week on the Internet |
15.0 (1-50) |
|
Game types* - poker - sports betting - casino - lottery - slot machines - scratching games - other |
99.7 19.0 7.1 3.9 3.5 3.2 1.9 |
|
Number of gaming sites visited last 3 months - only one - more than one |
42.2 57.8 |
|
In the last 12 months gambling has been: - my main source of income - an ancillary source of income - not a significant source of income |
14.5 30.2 55.3 |
* Total exceeds 100% due to multiple responses
Comment: Is it possible to include the questionnaire/questions as an Appendix here?
Answer: The questions specifically made for the survey could be added in form of appendix
Comment: Discussion: Study limitations needs more detail; the authors should consider at least limitations of the cross-sectional nature of the study and limitations related to the problem gambling measure used.
Answer: Accordingly these limitations were added in the discussion
Comment: Terminology issues: Overall, I think that the terminology used in this paper is not uniform and is confusing. Sometimes the authors use the term “Internet poker” and sometimes (see e.g. Table 2) “online poker”. Sometimes the authors use the term RG measures and sometimes (gambling) prevention measures. There are several terminology-related issues to be re-considered. Also in the abstract, the authors use the term “gambling prevention strategies”. However, I think that most of the offered strategies were planned to reduce (or manage excessive) gambling, not prevent all gambling. Generally speaking, gambling prevention strategies include several types of strategies from societal level interventions to tools offered to individual gamblers etc..
Answer 1:
Thank you for your helpful comment
In order to resolve the main terminology issue, according to the answer given to your 3rd comment we choose to replace the “prevention strategies” term by Harm-reduction, in the title, text and abstract.
Answer 2: Online" has come to describe activities performed on and data available on the Internet (Dictionary of British and World English. Oxford University Press) « Being online means that the equipment or subsystem is connected, or that it is ready for use. "Online" has come to describe activities performed on and data available on the Internet, for example: "online gambling".
Comment: Discussion: What about those people at risk of gambling problems? From the public health perspective, strategies for managing gambling and to prevent harm among this lower risk group of gamblers are also very important.
Answer:
Most of the HR strategies reported in the present study were endorsed by the participants across different levels of gambling severity suggesting possible usefulness also among the lower risk group of gamblers. The effectiveness of such interventions is however limited by the extent to which gamblers are prone to use the systems when they need it. HR strategies could be further improved by better awareness of emotional reactivity during game sessions and their possible impact on players’ loss of control64 In addition to effective HR tools, other preventative population approach such as supply reduction are needed to reduce the global burden of problem Internet gambling 12 .
The following reference were added:
McMahon N, Thomson K, Kaner E, Bambra C. Effects of prevention and harm reduction interventions on gambling behaviours and gambling related harm: An umbrella review. Addictive behaviors 2019;90:380-8.
Moreau A, Sevigny S, Giroux I, Chauchard E. Ability to Discriminate Online Poker Tilt Episodes: A New Way to Prevent Excessive Gambling? Journal of gambling studies / co-sponsored by the National Council on Problem Gambling and Institute for the Study of Gambling and Commercial Gaming 2020;36:699-711.
Submission Date
27 August 2020
Appendix:
Questions specifically written for the survey
|
|
|
Age |
|
Sex - male |
|
Living status: I live - alone - in a relationship - with my family - with friends - with roommates |
|
Annual income (in euros) - less than 30,000 - between 30,000 and 50,000 - more than 50,000 |
|
I've been gambling on the Internet since: - less than 2 years - between 2 and 5 years - more than 5 years |
|
Number of hours gambling in a week on the Internet : |
|
On the Internet, which game(s) do you play? (Several answers possible): - poker - sports betting - casino - lottery - slot machines - scratching games - other |
|
Number of gaming sites visited last 3 months - only one - more than one |
|
In the last 12 months gambling has been: - my main source of income - an ancillary source of income - not a significant source of income |
|
Did you borrow money in the past twelve months to gamble or to pay for gambling debts? - yes . No |
|
For a small number of players, gambling may become excessive, i.e. it is difficult for them to control their gambling time and/or the amount of money spent. How do you think these people could be helped to control or prevent this problem? Please rate each strategy according to the following answer options: 1) this wouldn’t help at all 2) this would be slightly helpful 3) this would be moderately helpful 4) this would be very helpful 5) this would extremely helpful |
|
1. information on excessive gambling and its consequences: |
|
2. information on specialized healthcare centres: |
|
3.Setting individual time limits on Internet gambling sites:
|
|
4. Messages and warnings based on play time limits:
|
|
5. Gambling opportunities are automatically blocked after a set time (mandatory limit): |
|
6. Setting individual money limits, for instance, players can set their own money limit to gamble and to lose on their profile:
|
|
7.Messages and warnings in relation to the total amount wagered and lost:
|
|
8.Gambling options are automatically blocked after a certain amount wagered and lost (mandatory limit) :
|
|
9.Offering online forums for discussion with other gamblers:
|
|
10. « online support», with a specialist on a website dedicated to gambling problems |

This manuscript is a resubmission of an earlier submission. The following is a list of the peer review reports and author responses from that submission.
Round 1
Reviewer 1 Report
The authors have generated an interesting set of data and use it to explore an important issue--strategies to best deal with harm minimization among gamblers. However, as written there are weaknesses in both presentation and methods that keep this paper from making a stronger contribution to the literature.
1. In places the writing style hinders understanding. For example, on P. 1, the first sentence of the Abstract reads "A part of online poker players present problem gambling." Some online poker players? Some specific percentage X of online poker players? Similarly, on P. 2, "land based gamblers" is ambiguous. I'm guessing that the authors mean live gambling or off-line gambling, but I shouldn't have to guess.
2. The persuasiveness of some claims made by the authors could be stronger. On P. 2 they make a good point--that broader taxonomies of on-line gambling (and probably gambling in general) are needed. But other than contrasting poker-only vs. poker plus players they don't seem to follow-up on that point. In a similar vein, in the literature review they cite studies as addressing a point, but then fail to give any details about what those studies have to say and how they inform this study. Specifically, "Several studies have examined players' attitudes toward limit setting tools [12, 21, 24]" They give some details on study 12 but say nothing about studies 21 and 24.
3. Who are these respondents? I'd like to know a little more about the data generation process. Even simple information, such as nationality, would help the reader understand the process that generated the data.
4. A key point of analysis is to contrast gamblers in debt to those who aren't. However, Table 1 seems problematic to me. Does the PGSI include an item that reads something like "Have you borrowed money or sold anything to gamble?" If so, then it seems kind of odd that in Table 1 the first row analyzes PGSI scores between the two groups while the second then examines response to "Did you borrow money in the past twelve months to gamble or to pay for gambling debts?" Similarly, Table 2 compares indebted to no debt players on the PGSI. But since the PGSI includes an item on debt shouldn't it be no surprise that players in debt have higher PGSI scores than those who are not in debt?
5. The authors do lots of independent tests. However, they don't seem to control the p-values for the fact that they are running multiple tests on the data. On P. 4 they noted that they were Bonferroni adjustments on residuals but they don't make a similar adjustment in their interpretation of p values. Multiple independent tests will inflate the chance of at least one Type I error for that set (or family) of tests. For example in Table 1 they explore a set of 10 items and identify Item 1 and Item 2 as statistically significant. But if they controlled for the family error rate by a Bonferroni adjustment (in this case .05/10 or .005 as the critical p value) they would only identify Item 1 as statistically significant.
6. On P. 7 they write "In Table 2, women represented less than 8% of indebted players." Given that the base rate of women in the entire study was slightly less than 6%, it seems like that's about what one would expect if sex was uncorrelated with indebtedness. But as it's written, I don't know if that is the point the authors are trying to make.
7. P. 8, Table 2, I've already mentioned the lack of independence due to the PGSI score including an item about borrowing money.
8. P. 9, Table 3. Here is another place where the authors might want to control for family-wise error rate via a Bonferroni adjustment to the critical p value. If they make that adjustment (.05/10 = .005) none of the relationships are statistically significant.
9. All the tables need to be reformatted, they were difficult to read.
10. If I counted correctly, there are 36 separate tests of statistical significance. Six are statistically significant at the usual p = .05 level. But several of these violate the assumption of independence (the items that are double-dipping on borrowing money / debt). And if the authors controlled for family-wise Type I error rates, they'd lose other statistically significant results. I point this out not because I want to reinforce the bias against negative results, but to suggest to the authors that an interesting perspective on this paper is the lack of differences between the various groups studied in this paper.
Reviewer 2 Report
Thank you very much for the opportunity to review this manuscript. This paper is clearly written and the analyses seem reasonable for the data used. I especially like the manuscript’s discussion of the unique challenges in targeting online poker players with RG strategies. I believe that the paper would be ready for publication after some minor corrections are made. I have outlined ways that I think would strengthen the manuscript below. Overall, the biggest weakness is in the presentation of the results but these issues would be easily remedied.
Introduction.
The introduction reads well and seems to give a generally good over view of limit setting and prevention messaging. I see that you have reference Ladoucer et al 2019. This piece might be more useful earlier in the introduction in framing what they identify as the 5 major types of RG strategies. A quick discussion of critiques of the RG paradigm in the introduction might be interesting but not altogether necessary (ex Hancock & Smith 2017).
Pg1 Provide an actual dollar estimate of the size of the industry
Pg 2 paragraph 3: responsible gambling has been abbreviated before it is introduced.
PG 2 paragraph 6: “social Responsibility” should probably be switched to responsible gambling
Style comment: the paper switches between past a present tense. The paper should be revised for consistent tense. Past tense is probably best.
Methods
Page 3 paragraph 6: the wording participants needs a little clarification on why participants were excluded
Section 2.2.1 the PGSI should indicate problem gambling as a score of 8 or higher. Not 8-9.
Results
Reporting of decimals for percentages should be consistent. Several use the French standard of a comma.
Provide total n for tables in the description. They appear to have different totals.
The percentages reported for men and women do not seem to track. Please double check this. A brief descriptive table of the sample would be helpful.
The tables should include the test statistic for comparisons made.
The formatting of the tables makes them difficult to read. Please reformat.
Discussion
For the most part the discussion seems reasonable and easy to interpret. I would suggest that the discussion problematizing responsible gambling strategies that is found in the discussion should be moved in the general discussion of the paper. These are issues for the field at large and less specific to the limitations of the research contained in this draft. Conversely, the discussion of the lack of women in the sample should be moved to the limitations.
“besides” is awkward here. “the study also examined…” might be better
Reviewer 3 Report
This study aims to examine online poker gamblers’ attitude towards prevention strategies. Comparisons were made between poker only and poker plus other games players. The manuscript was well-written. However, I have doubt on the significance and conceptual implications of comparing poker only and poker plus other games players. The authors need to provide a convincing reason.
Among 584 respondents, 311 were retained. What are the criteria to rule out the unqualified respondents?
Were the preventive measures borrowed from previous studies? Please provide the support.
I cannot see the necessity to merge the items, even though the sample size is small as some analyses should be able to analyze data with small sample size.
For the indebtedness item, it seems there is double-barrel issue. Someone may borrow money to gamble but not borrow money to pay for gambling debts.